# Enhancing east-west interface security in heterogeneous SDN *via* blockchain

Hamad Alrashede[1], Fathy Eassa[1], Abdullah Marish Ali[1],
Hosam Aljihani[2] and Faisal Albalwy[3]

[1] Department of Computer Science, Faculty of Computing and Information Technology, King Abdul Aziz University, Jeddah, Saudi Arabia
[2] Department of Computer Science, College of Computer Science and Engineering, Taibah University, Madinah, Saudi Arabia
[3] Department of Cybersecurity, College of Computer Science and Engineering, Taibah University, Madinah, Saudi Arabia



## ABSTRACT

Software defined networking (SDN) increasingly integrates multiple controllers from diverse vendors to enhance network scalability, flexibility, and reliability. However, such heterogeneous deployments pose significant security threats, especially at the east-west interface which is connecting these controllers. Existing solutions are inadequate for ensuring robust protection across multi-vendor SDN environments as most of them are meant to a specific type of attacks, use centralized solution, or designed for homogeneous SDN environments. This study proposes a blockchain-based security framework to address existing security gaps within heterogeneous SDN environments. The framework establishes a decentralized, robust, and interoperable security layer for distributed SDN controllers. By utilizing the Ethereum blockchain with customized smart contract-based checks, the proposed approach enables mutual authentication among controllers, secures data exchange, and controls network access. The framework effectively mitigates common SDN threats such as distributed denial-of-service (DDoS), man-in-the-middle (MitM), false data injection, and unauthorized access. Experimental results highlight the practicality of the solution, achieving a stable throughput of approximately 20 transactions per second with an average authentication latency of 28–40 ms. These results demonstrate that the proposed framework not only enhances inter-controller communication security but also maintains the network performance, making it a reliable and scalable solution for real-world SDN deployments.

# INTRODUCTION

Software defined networking (SDN) has come to the front line as a recent network architecture by allowing centralized control, programmable, and virtual network devices (*Turner et al., 2023*). With SDN, the control plane is thoroughly separated from the network devices (data plane). This enhancement grants network administrators more flexibility to manage, configure, and optimize the network resources (*Rahouti et al., 2022*). In 2023, the global SDN market is valued at USD 34.29 billion, and it is expected to grow at

Corresponding author
Hamad Alrashede,
algiladi@gmail.com

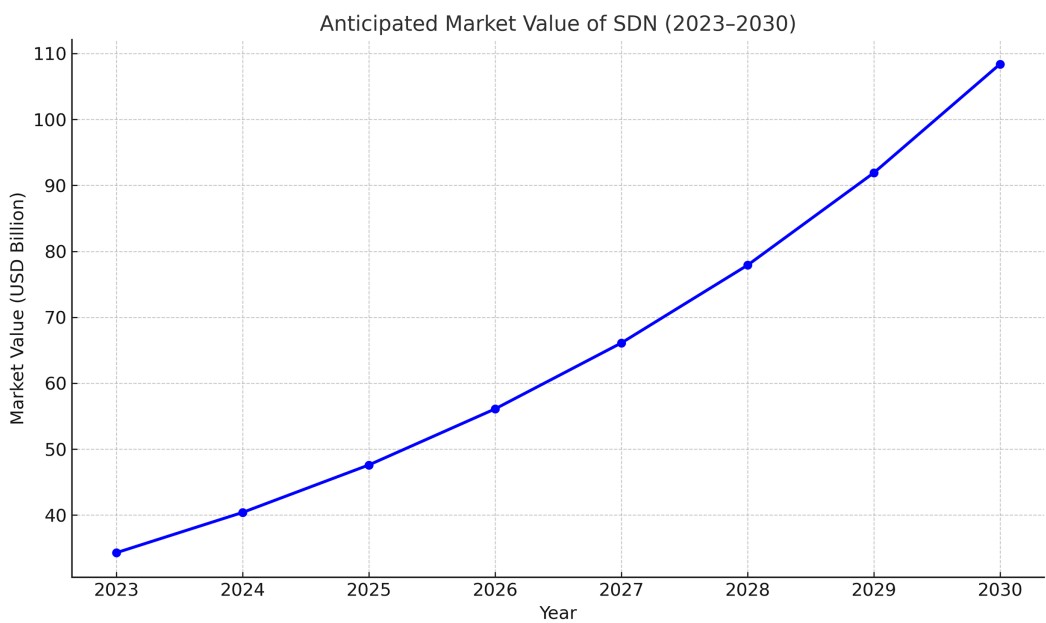

**Figure 1  Anticipated market value of SDN in 2023–2030.**

a compound annual growth rate of 17.9% between 2024 and 2030 (*Grand View Research, 2023*); Fig. 1 illustrates this growth.

However, as SDN gains attraction across various industry sectors, the integration of controllers from multiple vendors is unavoidable, leading to diverse SDN landscapes. These landscapes, while advantageous for vendor adaptability and feature fullness, encounter big obstacles concerning interoperability and security (*Farooq, Riaz & Alvi, 2023*). In these diverse SDN landscapes, the east-west interface plays a main role in facilitating communication among controllers, this interface is essential for supporting network scalability, as it empowers distributed controllers to exchange information, synchronize states, and guarantee the seamless operation of the entire network. Nevertheless, the absence of standardized security protocols among different controllers causes various vulnerabilities, positioning the east-west interface as a prime target for threats such as man-in-the-middle attacks, rogue controller infiltration, and data manipulation (*Maleh et al., 2023*). Most of previous research in securing east-west interface of SDN focuses on addressing specific threats such as false data injection (FDI), and distributed denial of service (DDoS). As well, they provide solution within homogeneous SDN controller environments, where controllers originate from a single vendor, thus simplifying security and interoperability concerns. However, real-world deployments increasingly adopt heterogeneous SDNs, incorporating controllers from multiple vendors such as Ryu, OpenDaylight, and ONOS. This diversity introduces significant challenges regarding interoperability, standardized security protocols, and centralized trust management, which remain inadequately explored by existing studies. Figure 2 presents the layered architecture of SDN, highlighting the east–west interface.

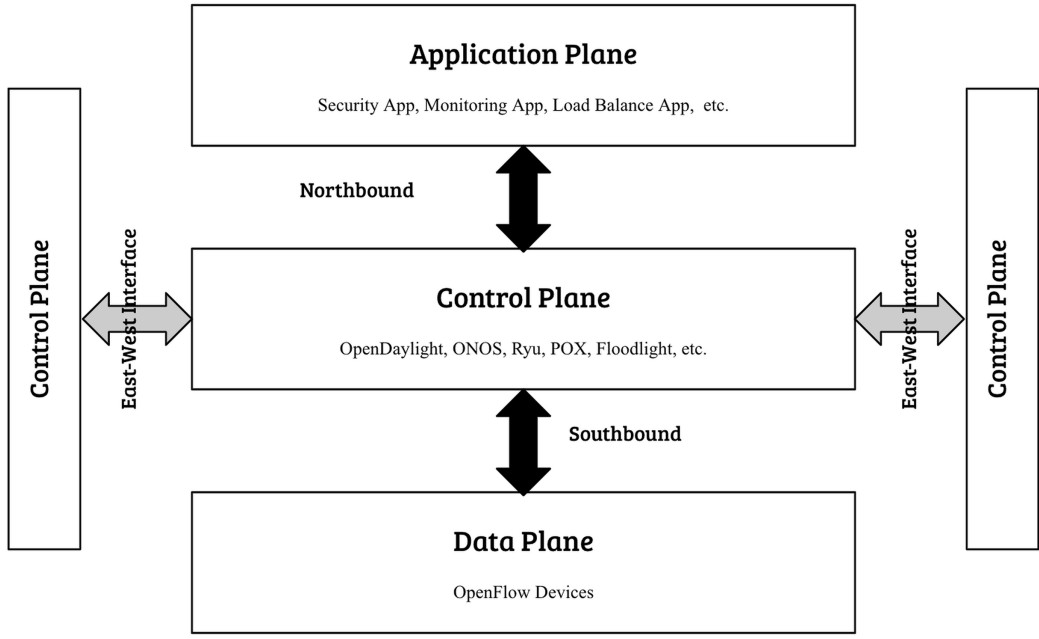

**Figure 2** SDN layered architecture including east-west interface.

To address these identified gaps, our study introduces a unique blockchain-based security framework explicitly tailored for heterogeneous SDN environments. In contrast to existing approaches, the proposed framework combines multiple critical security modules including mutual authentication, encryption/decryption, and decentralized access control —into a cohesive and unified solution. Its decentralized nature enhances resilience against traditional threats targeting centralized security architectures, representing a significant advancement in ensuring secure and interoperable communications among heterogeneous SDN controllers.

Blockchain technology is specifically chosen for securing east-west interface communication due to its inherent decentralization, immutability, and transparency. Traditional centralized security approaches are insufficient for heterogeneous SDN deployments, as centralized systems inherently possess single points of failure and trust vulnerabilities. Blockchain's decentralized ledger, coupled with smart-contract capabilities, provides a robust, trustless environment suitable for securely authenticating controllers, securely exchanging data, and enforcing strict access control without relying on centralized authorities.

This study presents the following key contributions:

- A secure and interoperable blockchain-based framework tailored for east-west interface in heterogeneous SDN environments.
- An analysis of the framework's resilience to common threats in east-west interface of heterogeneous SDN environments.

- A performance evaluation of blockchain integration within heterogeneous SDN environments.

The remainder of the article is organized as follows: "Related Work" reviews related literature. "Proposed Solution" presents the proposed blockchain-based security framework. Implementation details are described in "Implementation". "Validation and Evaluation" provides validation and performance evaluation. "Discussion and Results Analysis" offers a detailed analysis and discussion of the results. Finally, "Conclusion and Future Work" concludes the article and outlines directions for future research.

## RELATED WORK

This section presents a systematic review of prior research on security and interoperability challenges within software defined networking (SDN), with a focus on east-west interface communication. The reviewed literature is categorized into five main themes: general SDN security, encryption-based mechanisms, blockchain-based solutions, interoperability and architecture-focused studies, and alternative emerging approaches. Table 1 provides a summary of these studies with their contributions and limitations.

### General SDN security and threat landscape

*Wang et al. (2023)* provide a comprehensive overview of security threats in SDN, analyzing attacks targeting controllers and reviewing existing system-level protections. The study highlights the limitations of SDN controllers in handling a growing range of threats. *Jaraba et al. (2024)* investigate the strengths of SDN in the network management while emphasizing its vulnerability to DDoS threats. The study evaluates the effectiveness of current mitigation techniques and compares the impact of DDoS attacks across SDN communication layers (northbound, southbound, and east-west), also considering controller performance based on hardware usage and response times.

### Encryption-based security mechanisms

*Ghaly & Abdullah (2021)* identify AES, RSA, and hybrid encryption algorithms as effective for securing control plane communication in distributed SDN environments. The hybrid approach is found to outperform RSA alone in both security and performance. However, the proposed method depends on a centralized security model, limiting its scalability in distributed SDNs.

### Blockchain-enabled security frameworks

Blockchain integration in SDN has emerged as a promising solution for decentralizing control and enhancing security. *Sibiya, Molefe & Nleya (2024)* propose the Controller-Block model, combining blockchain with a density-based block structure and P2P networking to eliminate single points of failure and secure communications in cloud-based SDNs. *Alkhamisi, Katib & Buhari (2024)* introduce a Blockchain-based controller security (BCS) mechanism to protect communication among multiple controllers in MC-SDNs. The framework demonstrates strong performance under various attack scenarios, though it does not fully address scalability concerns in large networks.

**Table 1 A summary of prior studies highlighting their key contributions and associated limitations.**

| Reference | Approach | Contributions | Limitations |
| --- | --- | --- | --- |
| Wang et al. (2023) | Analysis of SDN security threats and built-in measures. | Highlights controller vulnerabilities and limitations against evolving threats. | Descriptive only; no novel solutions proposed. |
| Jaraba et al. (2024) | Evaluation of DDoS attacks across SDN communication layers. | Assesses DDoS impacts and evaluates controller-based mitigation based on hardware and latency. | Focused solely on DDoS; other threats not explored. |
| Moeyersons et al. (2020) | DOMINO framework for managing heterogeneous SDNs. | Supports pluggable SDN architecture across vendors. | Security aspects are not addressed. |
| Alrashede et al. (2024) | Blockchain-based security for east-west SDN interface. | Secures east-west communication in homogeneous SDNs. | Inapplicable to heterogeneous environments. |
| Almadani, Beg & Mahmoud (2021) | Distributing SDN Control Plane structure (DSF). | Synchronizes multiple controller architectures (flat, hierarchical, T-model). | Lacks integrated security features. |
| Hoang et al. (2022) | SINA for interoperability across SDN domains. | Improves scalability and consistency between controllers. | Does not include east-west security mechanisms. |
| Lam et al. (2016) | Identity-Based Cryptography (IBC) in SDN. | Simplifies key management for large-scale deployments. | Does not fully address system-wide security. |
| Moatemri et al. (2022) | Study on authentication delays in SDN. | Highlights performance differences across architectures. | Security focus limited to authentication. |
| Bülbül et al. (2022) | (TES) Trust Enhanced Security for routing. | Detects and isolates compromised switches to secure east-west paths. | Narrow focus on switch-level threats. |
| Derhab et al. (2021) | Blockchain-based multi-controller architecture (BMC-SDN). | Prevents false data injection using reputation metrics. | Does not address other major threats (e.g., MitM, DDoS). |
| Tollefson (2018) | Blockchain-enabled east-west interface for federated SDNs. | Promotes decentralized controller coordination. | Lacks comprehensive security evaluation. |
| Rahman et al. (2023) | Blockchain-SDN framework for secure IoT cloud computing. | Demonstrates blockchain adaptability for SDN. | Scope limited to IoT; lacks holistic security design. |
| Eltaief, Thabet & Kamel Ali (2022) | SDN with Multi-controllers using reputation-based blockchain. | Prevents false rule injections using trust mechanisms. | Focuses on one threat; lacks broader validation. |
| Nguyen et al. (2022) | Blockchain for east-west latency and gas optimization. | Enhances east-west communication efficiency. | Performance-oriented; overlooks security concerns. |
| Fan et al. (2021) | Blockchain-based coordination in distributed SDNs. | Enables secure inter-controller data exchange. | Ignores other security dimensions. |
| Boukria, Guerroumi & Romdhani (2019) | Blockchain-Based Controller for Flow Rule Integrity (BCFR). | Mitigates false flow rule injection. | Focused on one type of attack; lacks broader scope. |
| Ghaly & Abdullah (2021) | Hybrid encryption for control plane protection. | Identifies AES, RSA, and hybrid methods for SDN security. | Centralized mechanism; limited to encryption. |
| Sibiya, Molefe & Nleya (2024) | Controller-Block model integrating blockchain and SDN. | Enhances privacy and resilience via P2P blockchain. | Evaluation restricted to cloud-based setups. |
| Das et al. (2024) | Blockchain integration for D2D communication in SDN. | Secures IoT communications using smart contracts. | Focused on IoT; scalability not fully tested. |
| Alkhamisi, Katib & Buhari (2024) | Blockchain-based Controller Security (BCS) in MC-SDN. | Secures inter-controller communication via immutable ledgers. | Scalability challenges not addressed. |
| Wang et al. (2022) | East-west interface for fixed/mobile SDNs in 5G. | Proof of concept for 5G slicing in SDN. | Focused on cellular use case; security aspects limited. |
| Mahdi & Abdullah (2022) | Hybrid quantum key distribution (QKD) protocols. | Explores advanced cryptographic solutions for key management. | Practical deployment remains complex and resource-intensive. |

*Das et al. (2024)* embed blockchain into SDN components to secure device-to-device (D2D) communication, using smart contracts to enforce authentication and automate tamper-resistant security policies. *Alrashede et al. (2024)* present a blockchain-based framework for east-west security in distributed SDNs. However, the solution is limited to homogeneous controller environments, lacking support for heterogeneous architectures. *Derhab et al. (2021)* propose BMC-SDN, a multi-controller architecture using a blockchain-backed reputation mechanism to detect false data injection attacks. While effective for that specific threat, it does not address other key issues like man-in-the-middle attacks or privilege escalation. *Tollefson (2018)* explore a federated SDN architecture secured *via* blockchain, showcasing the benefits of decentralized management but without providing a comprehensive solution for east-west security. *Nguyen et al. (2022)* focus gas consumption and how to reduce latency in the SDN east-west interface through blockchain but neglect associated security concerns. *Fan et al. (2021)* propose a blockchain-based coordination method for distributed control planes, ensuring secure inter-controller communication but lacking broader threat coverage. *Boukria, Guerroumi & Romdhani (2019)* introduce a blockchain-based controller to prevent false injection for rules (BCFR), but the approach does not encompass other attack vectors.

## Architectural and interoperability solutions

Several works aim to improve SDN controller interoperability, though often without robust security integration. *Moeyersons et al. (2020)* introduce DOMINO, a pluggable framework for managing heterogeneous SDNs, yet security is not addressed. *Almadani, Beg & Mahmoud (2021)* present the distributed SDN control plane framework (DSF) to enhance synchronization across varied controller architectures (flat, hierarchical, T-model). However, DSF lacks built-in security mechanisms for multi-controller deployments. *Hoang et al. (2022)* develop the SINA framework to enable interoperability between distributed SDN domains, improving scalability and consistency but omitting east-west security considerations. *Lam et al. (2016)* utilize identity-based cryptography (IBC) to secure communication between SDN domains. While simplifying key management, this method may face performance and scalability challenges in large networks. *Moatemri et al. (2022)* conduct a comparative study on authentication delays across flat and hierarchical SDN architectures, offering performance insights but not addressing broader security concerns.

## Trust-based and alternative security approaches

*Bülbül et al. (2022)* propose the Trust Enhanced Security (TES) model, which focuses on detecting and isolating compromised switches in the east-west interface to ensure secure routing. *Eltaief, Thabet & Kamel Ali (2022)* combine blockchain and trust mechanisms in a multi-controller SDN environment to prevent false flow rule injection. However, the proposed solution addresses only one specific threat and lacks validation in heterogeneous, real-world scenarios. *Rahman et al. (2023)* develop a blockchain-SDN architecture tailored to secure industrial IoT environments. While it leverages blockchain's strengths, its scope is limited to IoT use cases and does not offer a comprehensive solution for east-west

communication. *Wang et al. (2022)* introduce an east-west interface framework supporting both fixed and mobile SDN controllers, providing a proof of concept for 5G slicing. However, security aspects are not deeply explored. *Mahdi & Abdullah (2022)* investigate hybrid quantum key distribution protocols as an innovative method for enhancing SDN security. Despite their promise, these techniques face practical deployment challenges due to complexity and high resource requirements.

### Research gap

While significant progress has been made in the literature for securing the east-west interface of SDN environments, much of the existing literature remains limited in scope. Most prior work either addresses specific security concerns, adopts centralized security approaches, or targets homogeneous SDN architectures without addressing the complexity of integrating controllers from different vendors. This gap highlights the need for a security framework that can handle diverse controller architectures and provide comprehensive security measures. The security and interoperability challenges that are associated with integrating controllers from different vendors have not been adequately explored, leaving the east-west interface vulnerable in heterogeneous environments. To bridge this gap, the present study introduces a blockchain-based security framework designed to facilitate secure and interoperable communication among heterogeneous SDN controllers. The proposed solution leverages blockchain's decentralized, and immutable nature to provide mutual authentication, data integrity, and access control, thereby enhancing the overall security and scalability of heterogeneous SDN deployments.

## PROPOSED SOLUTION

This section thoroughly discusses and examines the proposed blockchain-based security framework aimed at enhancing the security of the east-west interface within SDN environments.

### Framework architecture

The structure of the suggested framework is incorporating blockchain technology into the communication system among different SDN controllers to guarantee secure and authenticated interactions that are resilient against various attack methods. The framework includes different security modules based on blockchain technology. In comparison to methods that focus solely on specific security elements like authentication and safeguarding against false data injections our proposed framework provides a holistic solution by merging mutual authentication with encryption/decryption processes, access control measures and decentralized trust mechanisms. Figure 3 displays the design of the suggested system that allows controllers from various suppliers like ONOS, OpenDaylight and Ryu to connect and communicate through a blockchain driven framework.

Every interaction of the controllers is recorded on the distributed ledger to guarantee transparency. Using the blockchain enables the framework to remove the dependence on centralized authority, which is a vulnerability often found in traditional computer networks and previous studies. This distributed method guarantees that no single

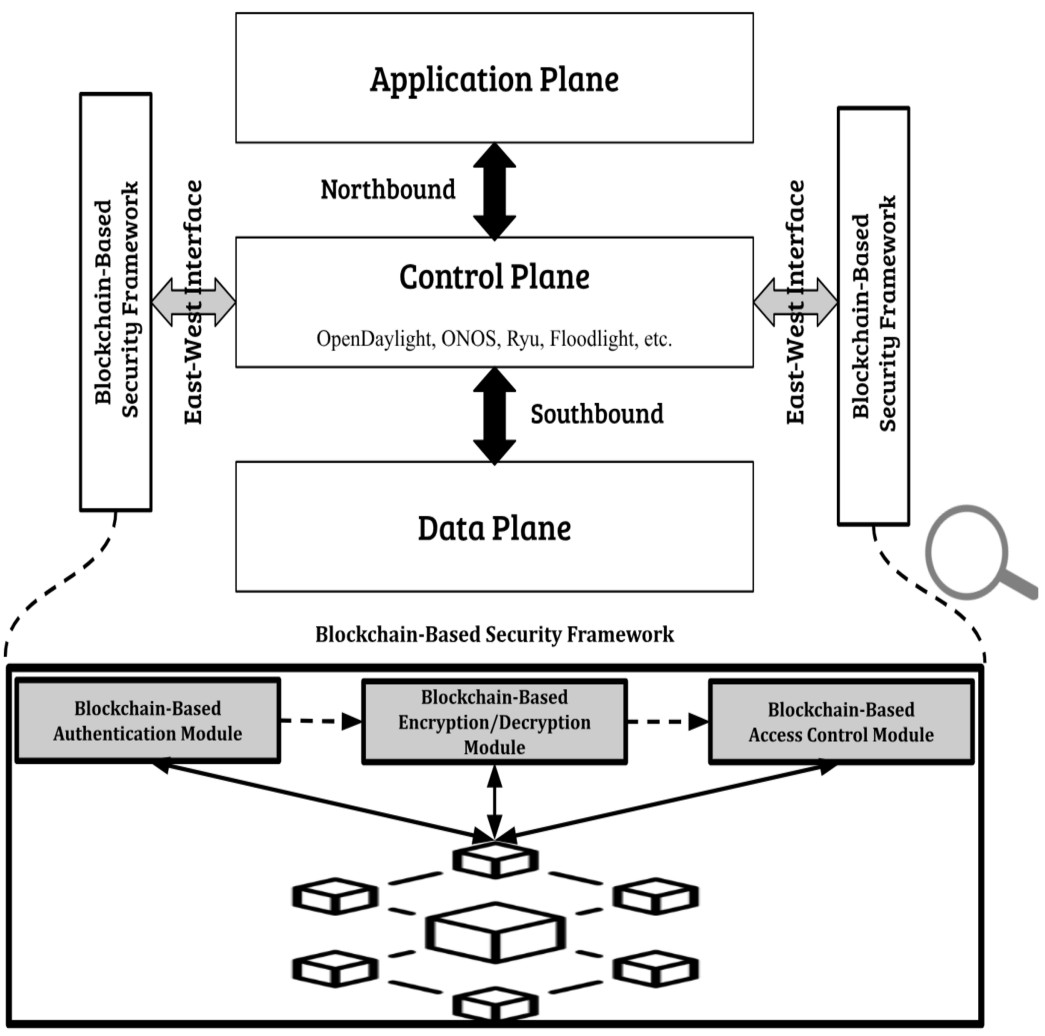

**Figure 3** **Proposed blockchain-based security framework.**

organization can undermine the system's security and strengthens the network's ability to face threats like DDoS, man in the middle attacks (MitMs), fake data injection and unauthorized access. The architecture is designed to scale seamlessly and efficiently, accommodating an increasing number of controllers without compromising performance or reliability.

## Framework unique features

The proposed security framework incorporates several distinctive features specifically designed to address security challenges associated with the east-west interface in heterogeneous SDN environments. The first notable feature is the decentralized trust mechanism. Leveraging blockchain's inherently decentralized structure, the framework establishes mutual trust among SDN controllers from various vendors, eliminating reliance on centralized authorities. Public keys and permission data for all controllers are

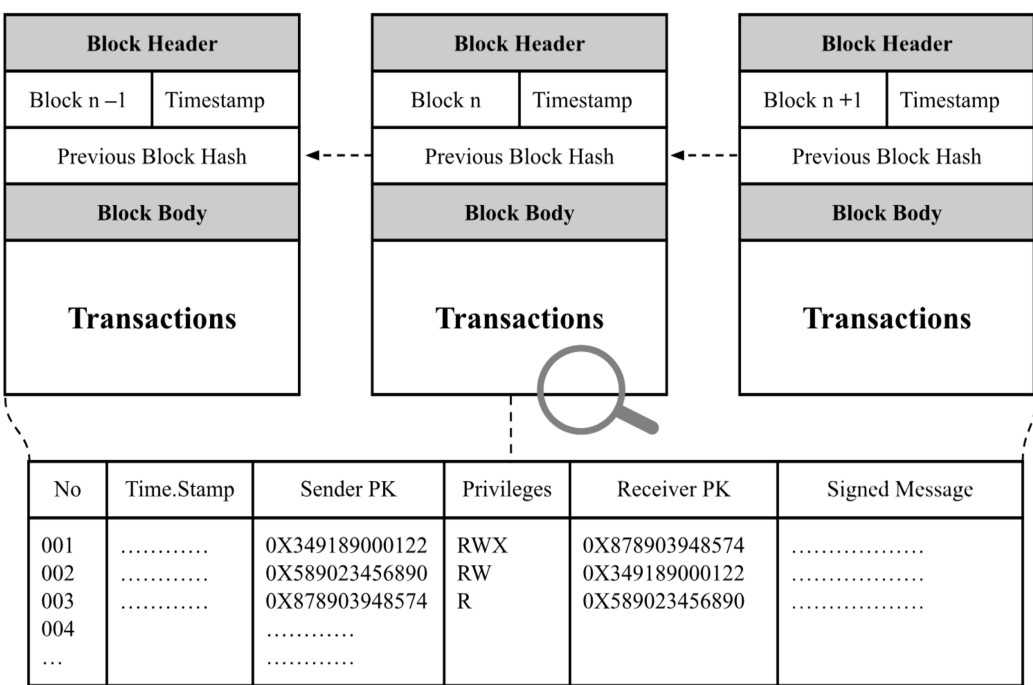

**Figure 4 Transactions recorded on the distributed ledger, comprising public keys, digitally signed messages, and access privileges.**

securely recorded on a distributed immutable ledger, effectively preventing unauthorized or malicious modifications. Additionally, every transaction among controllers is transparently logged, ensuring a secure and auditable record of interactions. Figure 4 illustrates the data structure used for transactions stored within this distributed ledger.

Another key feature of the framework is the authentication module, which ensures that only verified controllers participate in network interactions. By retrieving and verifying controllers' public keys directly from the blockchain, the module authenticates controllers before enabling communication, effectively preventing unauthorized access. The framework further strengthens secure interactions through its encryption/decryption module. Each message transmitted between controllers is digitally signed using the sender's private key and encrypted with the recipient's public key, providing robust protection against interception and tampering. These encrypted messages are securely stored on the blockchain, ensuring that only the intended recipient can decrypt the communication using their private key.

The access control module is another unique feature of the proposed framework, which assigns permission levels to each controller during the registration phase under the supervision of the network administrator, and securely logs them onto the distributed ledger. Access privileges are regulated using blacklists and whitelists that are stored in the blockchain to ensure that only authorized controllers can access data or carry out essential tasks. This module plays a role in preventing unauthorized access and privilege escalation in the diverse SDN environment while supporting overall security measures.

## Framework phases

### Registration phase

The SDN network administrator deploys Solidity smart contracts onto the blockchain to specify the criteria required for network membership and participation. When a new controller requests to join the network, the smart contract automatically checks whether the controller meets the established criteria and then notifies the administrator. Subsequently, the administrator reviews the registration request and decides whether to approve or reject it. Once approved, the controller's public key is recorded on the blockchain and designated as trusted. Additionally, during the registration stage, the network administrator assigns specific permission levels to each controller. Figure 5 presents a flowchart illustrating the steps involved in this registration process.

### Authentication phase

The registered controllers leverage the blockchain to perform mutual authentication. Initially, the sending controller queries the blockchain to retrieve the latest information regarding the intended recipient, then initiates a communication request *via* the blockchain. Subsequently, the receiving controller consults the blockchain to ensure the sender is recognized as a trusted entity. Only upon successful verification of the sender's identity does the receiving controller authorize communication through the blockchain. This mechanism ensures secure and authenticated interactions among controllers within the network, as illustrated in Fig. 6.

### Encryption/decryption phase

To protect communications between registered controllers, messages are exchanged using blockchain transactions. Initially, the sender digitally signs the message with its private key and encrypts it using the recipient's public key. The encrypted message is then transmitted as a transaction on the blockchain and broadcast across the entire network, after which the recipient is alerted about the new incoming message. Upon receiving this notification, the recipient retrieves the blockchain transaction, decrypts the message with its private key, and verifies the authenticity by validating the sender's digital signature against the corresponding public key. This secure layered method guarantees that only the intended recipient can access the message content, and verifies that the message indeed originates from the claimed sender. The detailed dataflow for this communication process is depicted in Fig. 7.

### Access control phase

The proposed framework facilitates access control by storing policies directly on the blockchain's immutable ledger. These policies involve straightforward mechanisms such as whitelisting and blacklisting based on controller public keys. Registered controllers must present valid credentials to gain access to sensitive blockchain data. According to their compliance with predefined conditions and rules, controllers' public keys are dynamically added to either the whitelist, permitting access, or the blacklist, restricting access. This strategy offers a secure and flexible mechanism for controlling data access across the network. The permission settings are recorded within the smart contract and can be

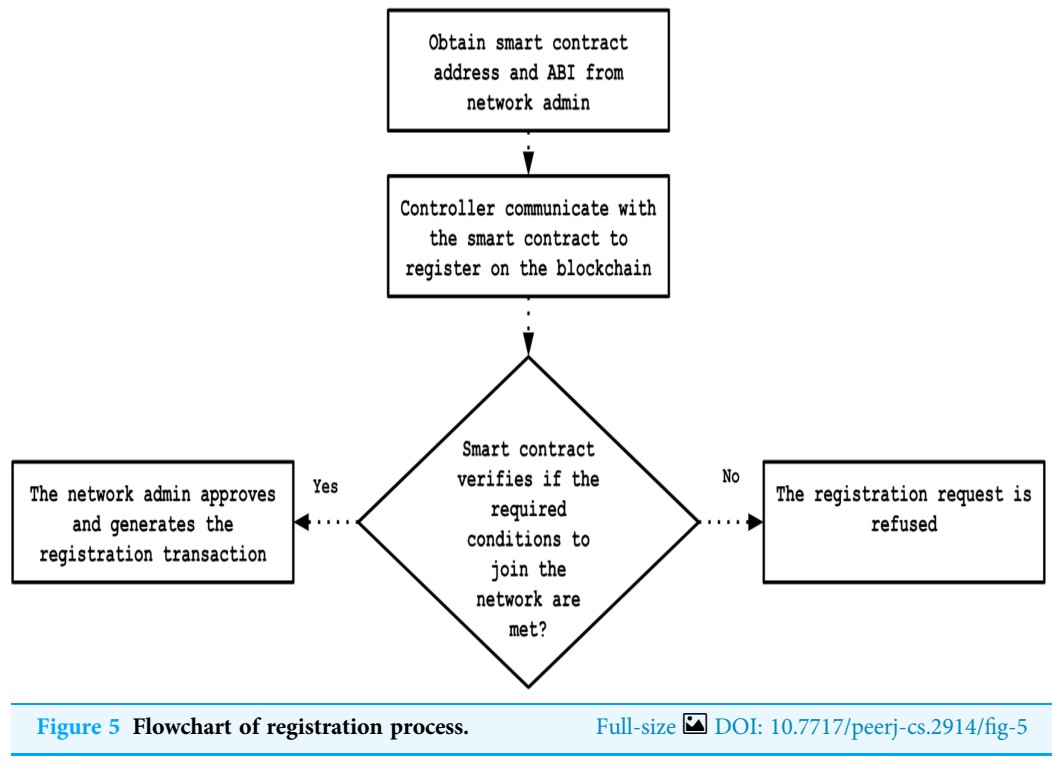

Figure 5 **Flowchart of registration process.**

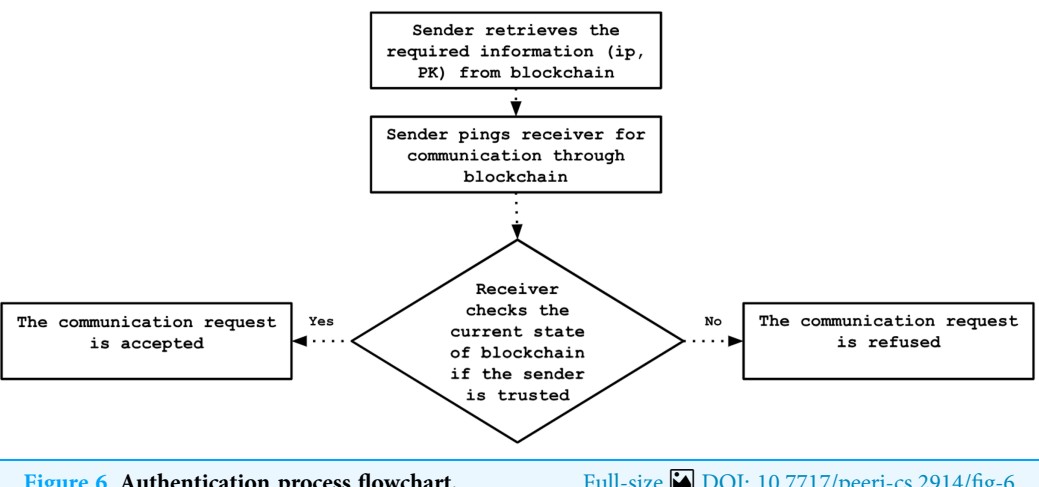

Figure 6 **Authentication process flowchart.**

dynamically updated in response to events like detecting malicious activity. The access control procedure is illustrated in Fig. 8.

## IMPLEMENTATION

In this section we provide an explanation of how the experiment was simulated and how the suggested security framework based on blockchain technology was configured. The source code of our study is available upon formal request, interested readers can access it through contacting the corresponding author.

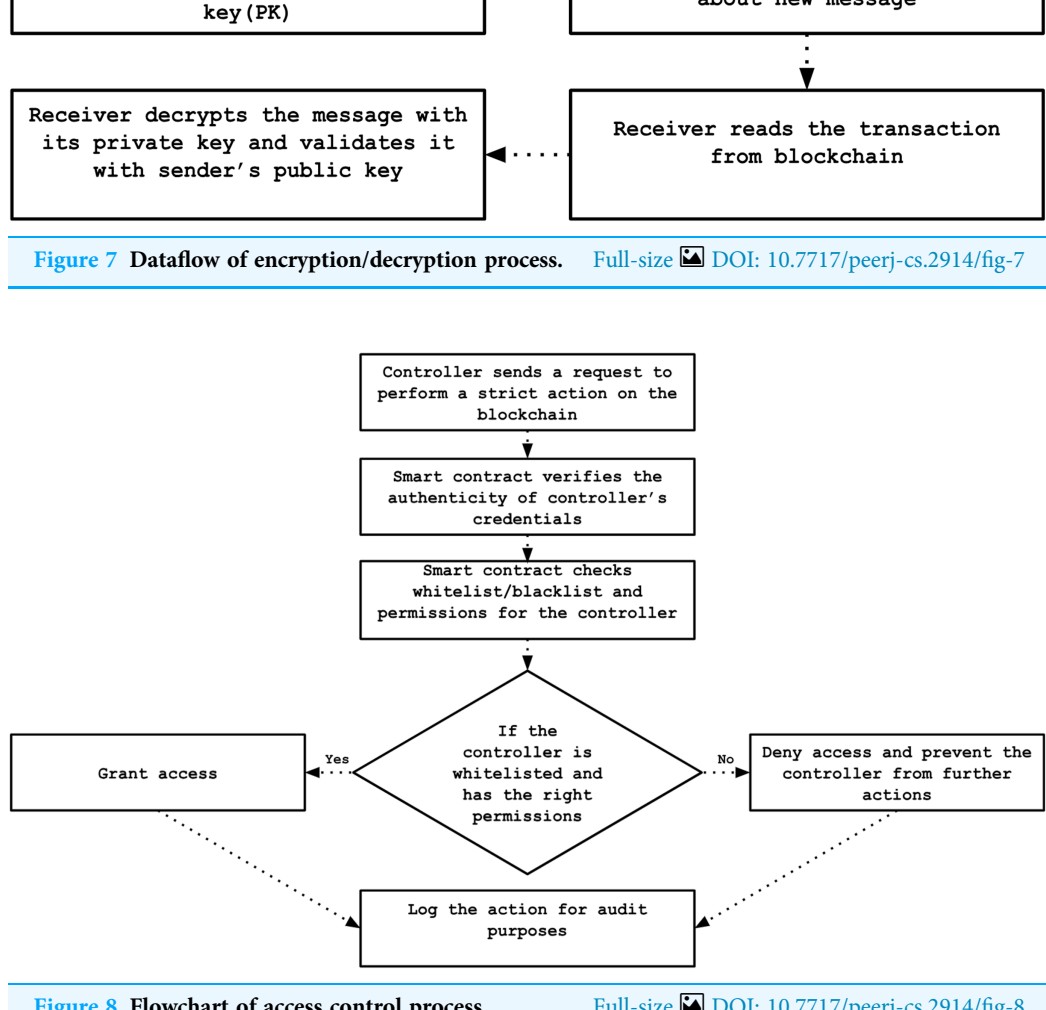

**Figure 7 Dataflow of encryption/decryption process.**

**Figure 8 Flowchart of access control process.**

## Experimental setup

Physical and virtual platforms (VMs) were used for performing the experiments, the used hardware and software are detailed in Tables 2 and 3.

The experimental environment consists of three distinct SDN topologies simulated in Mininet, each managed by a different SDN controller (Ryu, OpenDaylight, ONOS). Each topology comprised of switches interconnected in a mesh configuration with redundant paths to mimic realistic deployment conditions. Specifically, each topology included two switches, each connected to two simulated hosts.

Ganache was used for the blockchain simulations, configured with standard Ethereum parameters: a block size of 1 MB, a gas limit of 6,721,975 per block, and a Proof-of-Authority (PoA) consensus mechanism to simulate controlled access in permissioned blockchain environments. The smart contract was programmed using Solidity v0.8.10,

**Table 2 Hardware and platforms used for environment simulation.**

| Hardware | Specification |
| --- | --- |
| Mac OS monterey | Intel Core i7, 2.9 GHz, 16 GB RAM |
| Windows 10 | Intel Core i7, 3.4 GHz, 16 GB RAM |
| Ubuntu Linux (Virtual machine) | vCPUs: 4, vRAM: 4 GB (VM hosted on Oracle VirtualBox 7.0.14) |

**Table 3 Environment simulation software.**

| Software | Version/Details |
| --- | --- |
| Ganache blockchain | v2.7.1—A virtual Ethereum blockchain used to simulate the environment |
| Mininet | v2.3.0—Network emulator for SDN infrastructure simulation |
| Ryu | v4.34—Python-based SDN controller for managing control plane |
| OpenDaylight | Java-based SDN controller for managing control plane |
| ONOS | Java-based SDN controller for managing control plane |
| Solidity | v0.8.10—Smart contract programming language used to write blockchain contracts. |
| Python | v3.10.12—Used to integrates python-based SDN controllers with blockchain |
| Java | v11.0.23—Used to integrates Java-based SDN controllers with blockchain |
| Web3.py | v16.18.0—Python API for interacting with the blockchain |
| Web3j | Java API for interacting with the blockchain |

with specific functions designed to enforce authentication and access control policies. The smart contract was deployed on Ganache *via* remix platform.

Ryu controller is a python-based controller; thus Web3.py API was used to connect it with blockchain. While OpenDaylight and ONOS are Java-based controllers, therefore Web3j API was utilized to connect them with blockchain. Each controller was registered on Ganache blockchain using the deployed smart contract and assigned unique public/private key pairs. The different controllers utilize the proposed framework scripts to retrieve the trusted public keys from the deployed smart contract and verify the credentials of each controller before initiating secure communication. The smart contract assists communication with the various security modules and ensures that all interactions are transacted and recorded on the blockchain for transparency.

Figure 9 illustrates the deployment architecture of our proposed blockchain-based security framework. SDN controllers (ONOS, Ryu, OpenDaylight) interact with the blockchain *via* customized APIs (Web3.py and Web3j). The deployed smart contract handles controllers registration, authentication, encryption/decryption processes, and access control, ensuring secure and authenticated interactions among heterogeneous controllers without centralized points of failure.

## Experimental simulation

- The simulation aims to assess the proposed blockchain-based security framework in securing inter-controller communication between the heterogeneous controllers in distributed SDN environments. The simulation process includes the following steps:
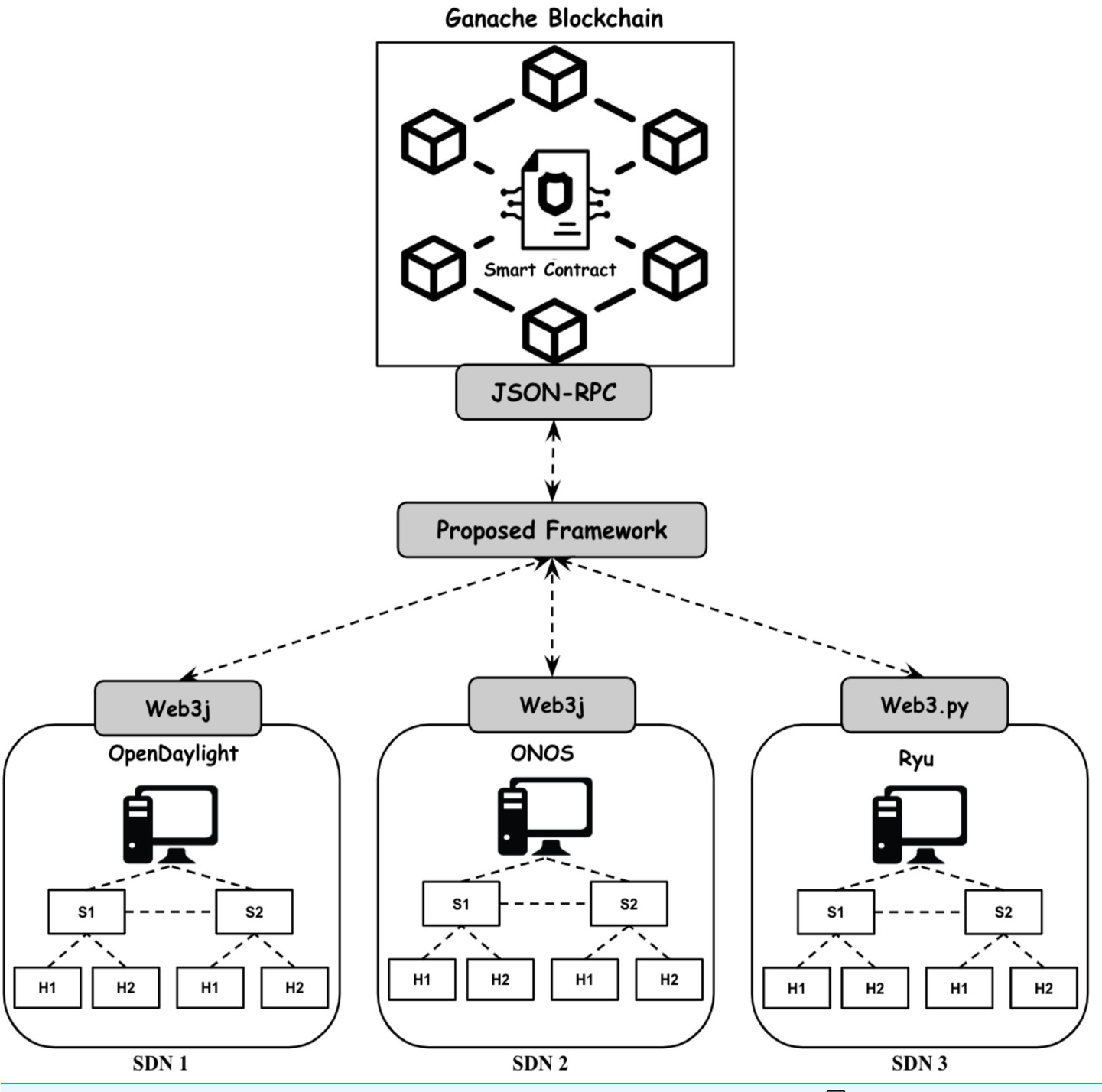

**Figure 9 Visual overview of experimental setups.**

- Initialization:
  - Private blockchain was initialized using Ganache.
  - Smart contract was deployed on Ganache using remix platform.

- Three mininet topologies were initialized and each one of them was connected to a separate controller on a separate virtual machine.

- Registration:

  - The heterogeneous controllers Ryu, OpenDaylight, and ONOS were registered on Ganache using the related API to interact with the deployed smart contract.

- Scenario creation:

The conducted experiments involved six key scenarios to evaluate the feasibility and robustness of the proposed security modules:

1. Legitimate communication request: Trusted controllers with valid credentials try to communicate and exchange messages.
2. Illegitimate communication request: Untrusted controllers or blacklisted ones try to communicate and exchange messages.
3. Unauthorized access attempts: Unauthorized controllers or controllers with no privileges attempting to access strict information on the blockchain.
4. False data injection attack: Injection of fraudulent transactions attempting to alter network state or flow tables.
5. Man-in-the-Middle (MitM): The communication channels between the connected controllers are intercepted to modify, replay, or spoof traffic.
6. Privileges escalation: An attacker attempts to elevate the permissions beyond what is authorized.

## VALIDATION AND EVALUATION

To validate and evaluate the proposed framework we conducted a series of experimental simulations designed to assess its effectiveness and performance. The produced data that related to the different conducted scenarios during the experiments were collected to provide insights into the framework's performance, and several metrics were taken into account. The metrics used for evaluation include the registration time, defined as the duration required to successfully register a controller on the blockchain; the authentication time, representing the period needed to verify controller credentials; the rate of false positives and negatives, measuring the frequency of incorrect approval or rejection of access requests; and the system throughput, indicating the number of successfully committed transactions (Ctx) processed per second.

### Validation

In validation assessment we focused on ensuring that the proposed framework could securely manage communication between the different heterogeneous controllers, and control the access into the blockchain-based SDN environment. The simulation results confirmed successful mutual authentication, encryption/decryption, and access control

between the heterogeneous controllers through Ganache blockchain, ensuring that only verified entities could communicate and participate in the network.

The proposed framework successfully prevented all the malicious activities, proving the reliability of the proposed system in maintaining a secure inter-controller communication through the east-west interface in the homogeneous and heterogeneous SDN environments. Table 4 highlights the common threat models in east-west interface communications and demonstrates how the proposed framework could prevent them.

## Evaluation

In evaluation assessment, the experiments were repeated across various scenarios to analyze the framework performance with focusing on key metrics like latency and throughput. The latency is the time taken for a transaction to be saved on the blockchain. The next Eq. (1) is used for calculating the latency.

$$Latency = CommitTime - SubmitTime. \tag{1}$$

The throughput is determined as the summation of committed transactions (Ctx) in a given period of time. Ctx represents the number of committed transactions, and (Tts) represents the total time in seconds. Equation (2) is used for calculating the Throughput.

$$Throughput = \sum Ctx/Tts. \tag{2}$$

The different experiments exposed variation in performance during the registration and authentication process, as well the overall system throughput. Figures 10–17 illustrate this variation, showcasing the registration time, authentication latency, and system throughput across diverse scenarios in diverse environments. Figures 10–12 display the registration time, minimum, average, and maximum times for various controllers on different platforms as the number of controllers goes up from 1 to 10. There seems to be a rise in registration time for all three controllers with an increase in the number of controllers which indicates a minor impact on performance due to scaling.

Comparison in Fig. 13 displays the average registration durations of Ryu and ONOS controllers in a mixed setup alongside OpenDaylight controllers. Ryu exhibits the longest registration time on average among the three controllers mentioned with ONOS following next and OpenDaylight registering the shortest time on average. The analysis underscores disparities in performance, across these controllers within a block-chain-based environment where OpenDaylight stands out for its more efficient registration process.

Figures 14 and 15 illustrate the authentication delay across an increasing number of controllers from 2 to 20, comparing homogeneous and heterogeneous configurations. In the homogeneous setup there is a slight upward trend in delay, with max, average, and min values close together, indicating some stability but a gradual increase as more controllers are added. In the heterogeneous setup, the delay values are slightly higher overall compared to the homogeneous case.

The system throughput is shown in Figs. 16, and 17 as the number of transactions increases from 1 to 100. In both data statistics, the throughput starts at a high rate but gradually stabilizes as transactions increase, indicating that the system handles

**Table 4 Effectiveness of the proposed security framework against common threat models targeting the east–west interface.**

| Threat model | Description | Prevention with proposed framework |
|---|---|---|
| Rogue controller | A malicious controller is registered among SDN environments. | The deployed smart contract manages the registration process for each single controller. |
| DDOS | The attacker compromises a huge number of devices and uses them to block the service from the network. | Initially, it is not allowed to communicate with the proposed network before registration on the permissioned blockchain. |
| False data injection | Incorrect data is injected to SDN by a compromised controller, such as fake flows or topology changes, to create conflicts. | Only registered and trusted controllers can send transactions to the blockchain. |
| Man-in-the-middle | The communication channels are intercepted to modify, replay, or spoof traffic. | The encryption-decryption module prevents such attacks. |
| Privileges escalation | An attacker attempts to elevate the permissions beyond what is authorized. | Privileges are securely saved on a distributed immutable ledger to prevent unauthorized changes. |

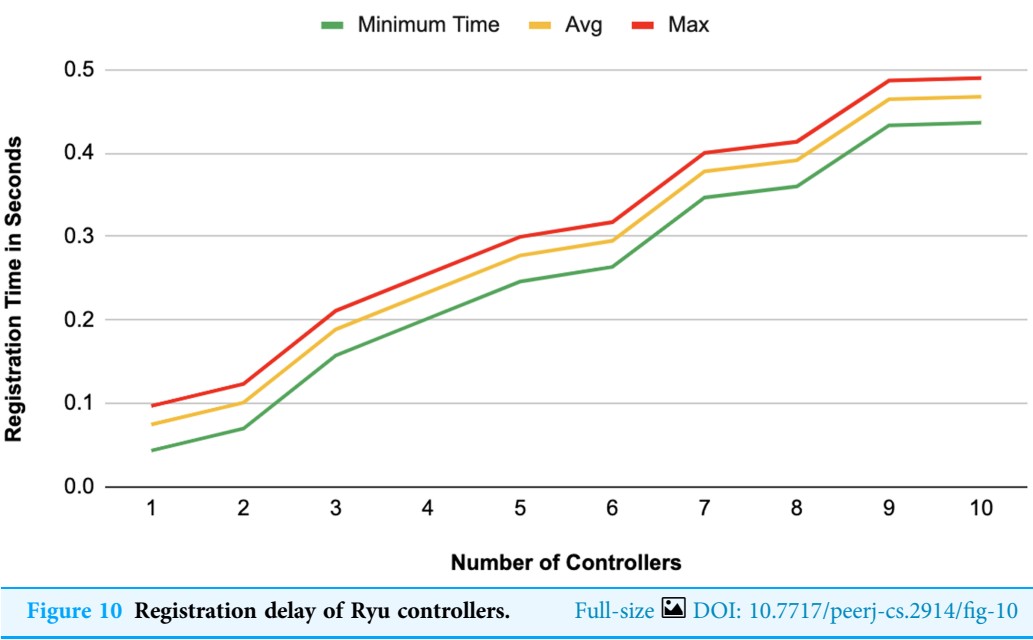

**Figure 10 Registration delay of Ryu controllers.** 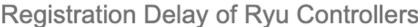

transactions efficiently over time. The time in seconds increases linearly, representing a consistent processing time per transaction. The homogeneous system has a slightly lower initial throughput than the heterogeneous system, suggesting that the heterogeneous systems may handle initial transactions better than the homogeneous ones.

## DISCUSSION AND RESULTS ANALYSIS

In this section, we study the results of our experiments, analyze the related impacts, and discuss the features of our proposed framework as compared to previous approaches in the

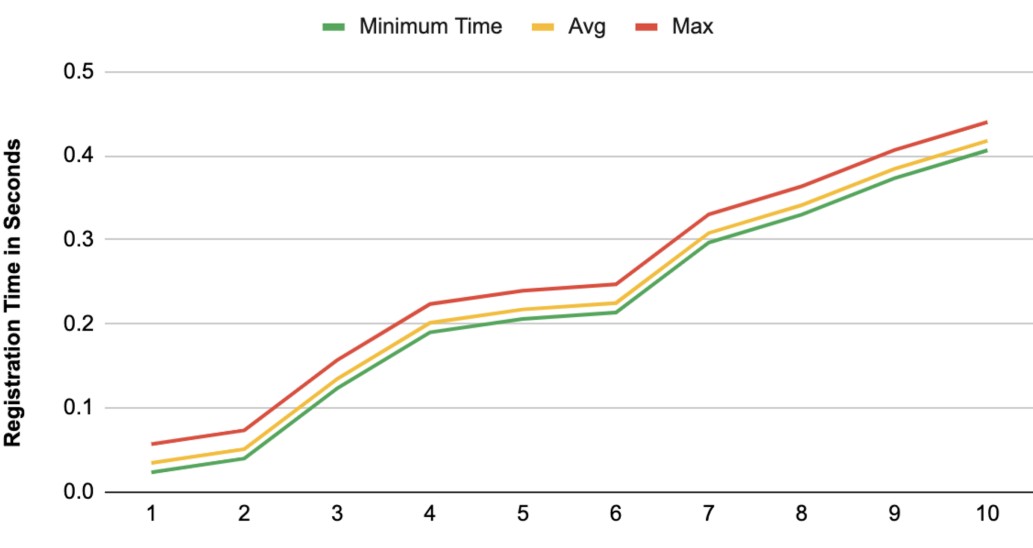

**Figure 11 Registration delay of OpenDaylight controllers.**

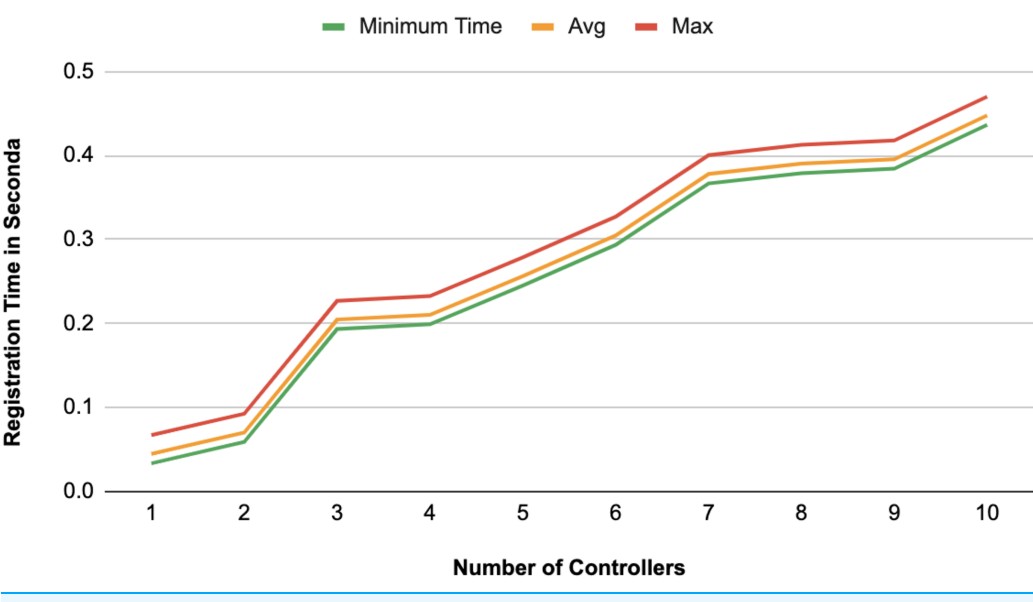

**Figure 12 Registration delay of ONOS controllers.**

literature. The section provides a comprehensive view of the effectiveness and contributions of our blockchain-based security framework for the east–west interface in heterogeneous SDN environments.

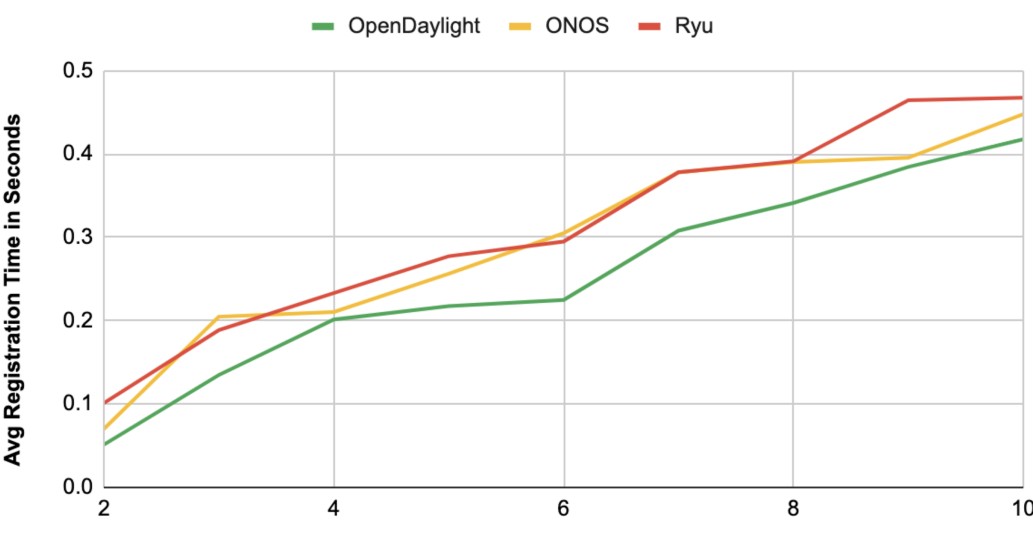

**Figure 13  Average registration delay of heterogeneous controllers.**

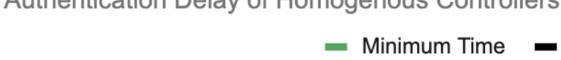

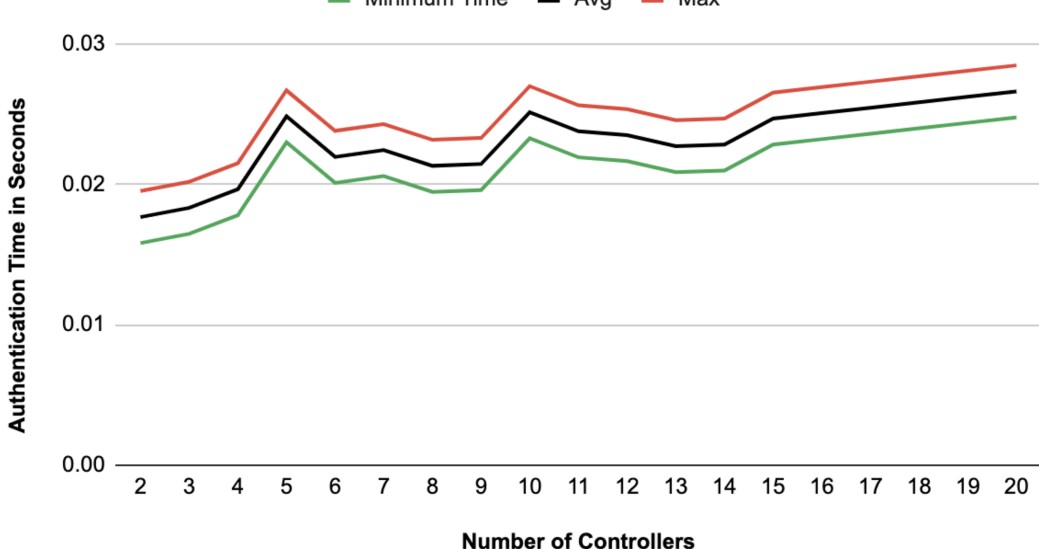

**Figure 14  Authentication delay of homogeneous controllers.**

## Results analysis

The experimental analysis of the proposed framework focuses on key performance metrics, including security robustness, registration latency, authentication latency, and system throughput. The results demonstrate that the framework maintains strong security while

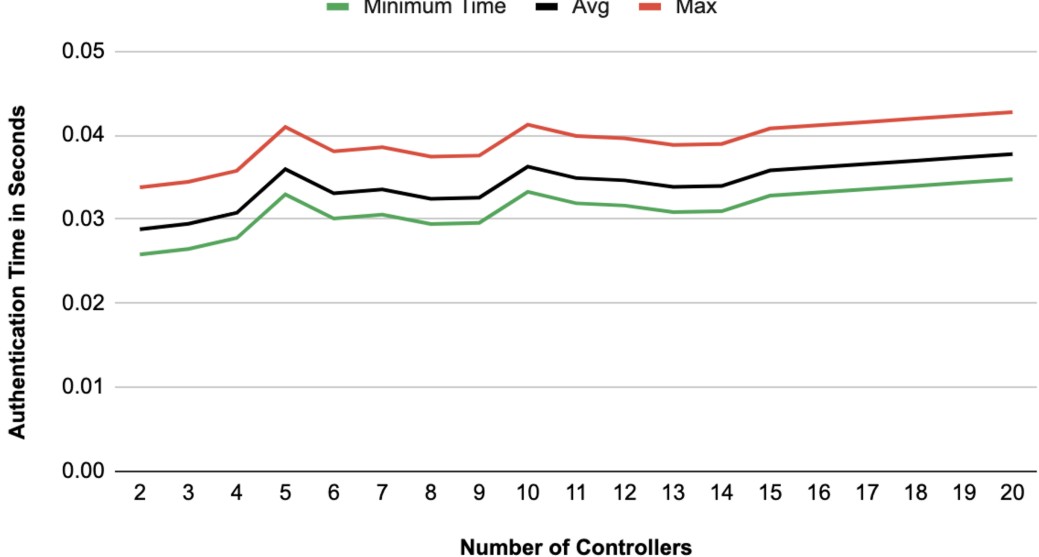

**Figure 15** Authentication delay of heterogeneous controllers.

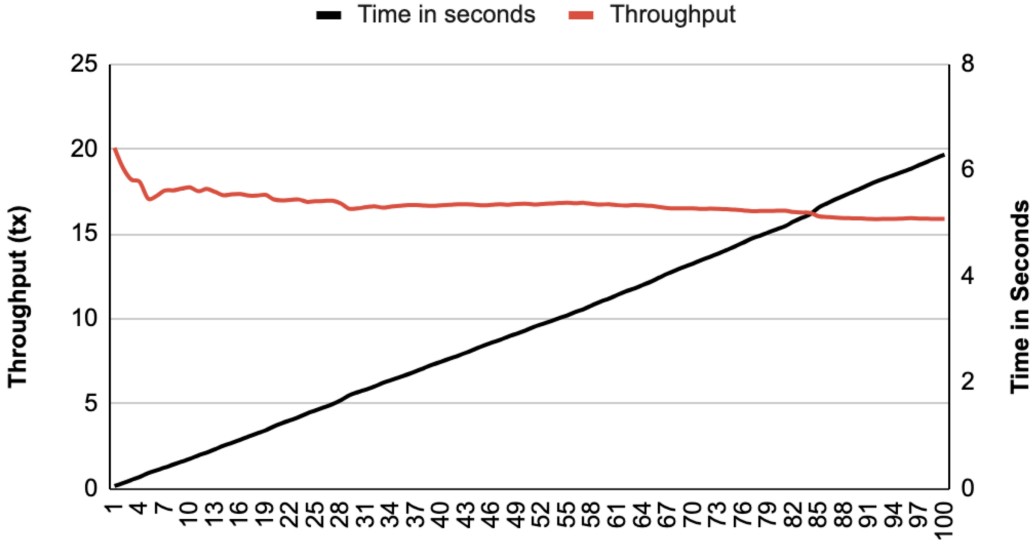

**Figure 16** Throughput of homogeneous controllers.

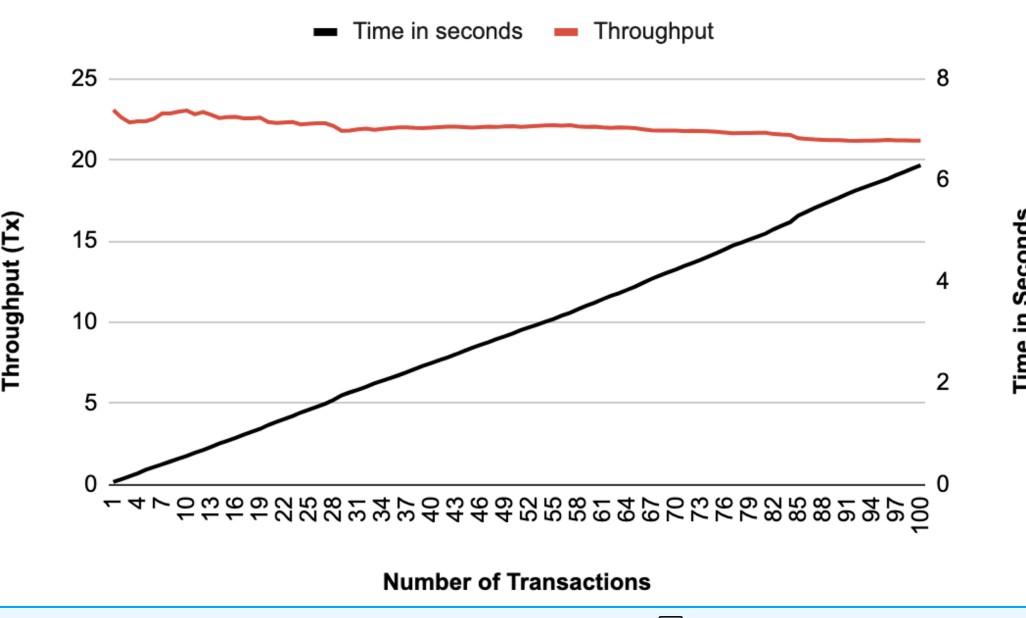

**Figure 17 Throughput of heterogeneous controllers.**

delivering efficient performance across various scenarios, with minimal impact on scalability as the system grows.

### Security evaluation

In our experimental scenarios we have tested the proposed framework against different types of common attacks on the east-west interface, including DDOS, false data injection, MitM, and unauthorized access. The use of blockchain to log every interaction and enforce mutual authentication ensured that all participating controllers were verified before communication was allowed. The proposed security framework successfully prevented all the malicious activities, proving the reliability of the proposed system in maintaining a secure inter-controller communication through the east-west interface in the homogeneous and heterogeneous SDN environments. The proposed solution stands out from existing techniques by addressing heterogeneous SDN controllers while offering a comprehensive defense against multiple security threats. In contrast, many existing approaches focus on a single security challenge or are limited to homogeneous network environments, making them less adaptable to diverse real-world deployments.

### Registration latency

On average, registering an additional controller on the blockchain-based SDN required approximately 0.10 s. This efficient registration process facilitates rapid controller integration into the network, thereby enhancing the scalability of the SDN environment. Notably, other related studies either omitted the registration phase entirely or did not report the associated time metrics.

### Authentication latency

The authentication process between homogeneous controllers was completed in an average of 0.020 s, whereas heterogeneous SDN environments exhibited a slightly higher average of approximately 0.030 s. These results indicate that homogeneous controller configurations provide better synchronization, with reducing authentication delays. When compared with related studies, such as *Derhab et al. (2021)* reporting 0.055 s, *Tollefson (2018)* with 0.068 s, and *Rahman et al. (2023)* achieving 0.021 s, our proposed solution demonstrates a clear advantage in terms of authentication performance.

### System throughput

The experimental results show that the proposed blockchain-based security framework achieved a stable throughput of about 20 transactions per second. This high rate means the system can process many transactions without slowing down, making it suitable for the large scale SDN environments. The throughput starts at a high rate and then becomes stable as the number of transactions increases, which demonstrates that the system manages transactions effectively over time. The time as well increases in a linear pattern, indicating a consistent processing time for each transaction. The homogeneous system initially has a slightly lower throughput as compared to the heterogeneous system, which means that heterogeneous SDN environments might handle initial transactions more efficiently.

## Comparative analysis

We have compared our proposed blockchain-based security framework to other related works by highlighting its unique features that are not present in similar research. Table 5 provides a qualitative comparison, showing how the proposed framework differs from the previous proposed schemes in many aspects.

## Study limitations

Since this study mainly focuses on a proof of concept (PoC), we used a blockchain simulation environment called Ganache blockchain, which may produce different results as compared to a live blockchain because of its built-in limitations. In our future research, we will involve testing on live blockchains to provide a more thorough analysis. As well, despite the promising achieved results, potential limitations exist concerning scalability and interoperability. Blockchain integration inherently introduces overhead, potentially impacting scalability as controller numbers increase. Interoperability between different vendor-specific controllers may pose practical challenges regarding the adaptation and integration of smart contracts. Further empirical studies in realistic large-scale environments are essential for accurately assessing and addressing these potential challenges.

**Table 5 Qualitative comparison between the proposed solution with previous approaches.**

| Reference | Decentralization | Multi-security | Scalability | Homogeneity | Heterogeneity |
|---|---|---|---|---|---|
| *Moeyersons et al. (2020)* | ✓ | ✗ | ✓ | ✓ | ✓ |
| *Alrashede et al. (2024)* | ✓ | ✓ | ✓ | ✓ | ✗ |
| *Almadani, Beg & Mahmoud (2021)* | ✓ | ✗ | ✗ | ✓ | ✗ |
| *Hoang et al. (2022)* | ✓ | ✗ | ✓ | ✓ | ✓ |
| *Moatemri et al. (2022)* | ✗ | ✓ | ✗ | ✓ | ✗ |
| *Bülbül et al. (2022)* | ✓ | ✓ | ✗ | ✗ | ✗ |
| *Derhab et al. (2021)* | ✓ | ✓ | ✗ | ✗ | ✗ |
| *Tollefson (2018)* | ✓ | ✗ | ✗ | ✓ | ✗ |
| *Rahman et al. (2023)* | ✓ | ✓ | ✓ | ✓ | ✗ |
| *Eltaief, Thabet & Kamel Ali (2022)* | ✗ | ✓ | ✗ | ✗ | ✗ |
| *Nguyen et al. (2022)* | ✓ | ✗ | ✗ | ✗ | ✗ |
| *Fan et al. (2021)* | ✓ | ✓ | ✗ | ✗ | ✗ |
| *Boukria, Guerroumi & Romdhani (2019)* | ✗ | ✓ | ✗ | ✓ | ✗ |
| Proposed solution | ✓ | ✓ | ✓ | ✓ | ✓ |

## CONCLUSION AND FUTURE WORK

In conclusion, this study introduced a novel security framework leveraging blockchain technology to safeguard communication within the east-west interfaces of heterogeneous SDN environments. The proposed framework comprises several security modules authentication, encryption/decryption, and access control, offering a robust and comprehensive solution for securing the east-west interface. To validate the feasibility of the solution, different controllers from various vendors, including Ryu, OpenDaylight, and ONOS, were integrated within a single SDN environment. Simulated experiments successfully mitigated common attacks on the east-west interface, such as man-in-the-middle attacks, unauthorized access, and false data injection, demonstrating the system's reliability in securing inter-controller communication in both homogeneous and heterogeneous SDN environments. The experimental results highlight the framework's effectiveness in enhancing inter-controller communication security while maintaining network performance. The average authentication latency for secure communication between heterogeneous controllers ranged from 28 to 40 ms, with a stable throughput of 20 transactions per second. This approach enables secure interactions among SDN controllers developed by different vendors, addressing a significant challenge in deploying heterogeneous SDN environments.

Future work will focus on real-world implementation to better understand practical hurdles and evaluate the system performance under actual traffic conditions. Additionally, integrating diverse blockchain platforms will provide a more comprehensive assessment of the proposed security framework.

### Funding

The authors received no funding for this work. The APC was supported by a grant (No. CRPG-00-0000) under the Cybersecurity Research and Innovation Pioneers Initiative, provided by the National Cybersecurity Authority (NCA) in the Kingdom of Saudi Arabia. The funders had no role in study design, data collection and analysis, decision to publish, or preparation of the manuscript.

### Grant Disclosures

The following grant information was disclosed by the authors:
APC: CRPG-00-0000.
Cybersecurity Research and Innovation Pioneers Initiative.
National Cybersecurity Authority (NCA) in the Kingdom of Saudi Arabia.

### Competing Interests

The authors declare no conflict of interest.

### Author Contributions

- Hamad Alrashede conceived and designed the experiments, performed the experiments, analyzed the data, performed the computation work, prepared figures and/or tables, authored or reviewed drafts of the article, and approved the final draft.
- Fathy Eassa analyzed the data, authored or reviewed drafts of the article, supervision, writing review, and editing, and approved the final draft.
- Abdullah Marish Ali analyzed the data, authored or reviewed drafts of the article, supervision, writing review, and editing, and approved the final draft.
- Hosam Aljihani conceived and designed the experiments, analyzed the data, authored or reviewed drafts of the article, writing review, and editing, and approved the final draft.
- Faisal Albalwy conceived and designed the experiments, analyzed the data, authored or reviewed drafts of the article, writing review, and editing, and approved the final draft.

### Data Availability

The code is available in the Supplemental Files.

### Supplemental Information

Supplemental information for this article can be found online at http://dx.doi.org/10.7717/peerj-cs.2914#supplemental-information.

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
