# Peer review of "Enhancing east-west interface security in heterogeneous SDN via blockchain"

_PeerJ Computer Science, doi:10.7717/peerj-cs.2914_

## Round 0.1 · original submission · Major Revisions

Please address all the comments of both reviewers.

Reviewer 1 ·

Basic reporting

While the introduction contextualizes the problem, it lacks a clear articulation of the research gap. The novelty and significance of the proposed solution are not sufficiently highlighted. The motivation for using blockchain technology specifically for east-west interface security should be better justified, emphasizing why existing solutions are inadequate.

Some figures are of low resolution and poorly formatted, making them difficult to interpret. They should be enhanced for better visibility and comprehension. For example , Fig. 1 and Fig 2.

Experimental design

The experimental setup lacks sufficient detail for replication. Key information is missing, including:
a) Network configurations and topologies used in the simulations.
b) Specific parameters for blockchain (e.g., block size, gas limits, and consensus algorithms).
c) Detailed descriptions of test scenarios and attack models (e.g., DDoS, false data injection).

The study uses Ganache, a private blockchain simulator, which raises questions about the generalizability of results to public or consortium blockchain networks. Testing on live blockchain platforms (e.g., Ethereum or Hyperledger) is strongly recommended.

Validity of the findings

Limitations are acknowledged but not sufficiently analyzed. The paper should discuss potential scalability issues, interoperability challenges with other SDN controllers, and network overhead introduced by blockchain integration.

Cite this review as

Reviewer 2 ·

Basic reporting

The authors addresses the security gaps in heterogeneous environments by proposing a blockchain-based security framework. The framework establishes a decentralized, robust, and interoperable security layer for distributed SDN controllers. By utilizing the Ethereum blockchain with customized smart contract-based checks, the proposed approach enables mutual authentication among controllers, secures data exchange, and controls network access. The framework eƻectively mitigates common SDN threats such as distributed denial-of-service (DDoS), man-in-the-middle (MitM), false data injection, and unauthorized access.

In the abstract, it is not mentioned why we need to use multiple controller.
What are the real motivations behind this solution?
The introduction section does not provide a systematic review of the existing approaches, motivations, and contribution of the proposed approach.
add some more relevant studies.

Experimental design

It needs to be improved to solve the problem discussed.
Figure 3 needs to be explained. How this new framework will be implemented.
In the proposed solution, how trust is computed? Which controller is considered as more trusted?
Most of the proposed solutions discuss the general mechanism of the proposed solution. How it can be deployed in real SDN?
Which types of problems of SDN controller can be considered to be solved using this solution?
The significance of the proposed system and how it is making a better impact than existing systems are not defined clearly.
The result and discussion section are so brief, as it does not explain the outcomes, how your approach is better than the existing work, or how your selected evaluation parameters are better than the existing approach parameters.

Validity of the findings

Reuslts need to be updated to show the real-time performance of the proposed solution.
You need to consider a case study to implement and evaluate the performance of the proposed solution.
The paper lacks the actual solution to the problem and its significant comparison with other techniques to detect different type of attacks. .

Additional comments

Figures should be added with more details.
Graphs should be added with more clarity and significance.

Cite this review as

---

## Round 0.2 · accepted · Accept

I am pleased to inform you that your work has now been accepted for publication in PeerJ Computer Science.

Please be advised that you cannot add or remove authors or references post-acceptance, regardless of the reviewers' request(s).

Thank you for submitting your work to this journal. I look forward to your continued contributions on behalf of the Editors of PeerJ Computer Science.

With kind regards,

Reviewer 1 ·

Basic reporting

no comment

Experimental design

no comment

Validity of the findings

no comment

Additional comments

Please make sure all figures have proper citation if required

Cite this review as

Reviewer 2 ·

Basic reporting

The improved version of the paper is good.

Experimental design

The authors has improved the design issues.

Validity of the findings

Now all the validation are ok.

Cite this review as